# Cell Therapy Based on Gingiva-Derived Mesenchymal Stem Cells Seeded in a Xenogeneic Collagen Matrix for Root Coverage of RT1 Gingival Lesions: An In Vivo Experimental Study

**DOI:** 10.3390/ijms23063248

**Published:** 2022-03-17

**Authors:** Nerea Sanchez, Fabio Vignoletti, Ignacio Sanz-Martin, Alejandro Coca, Javier Nuñez, Estela Maldonado, Javier Sanz-Esporrin, Irene Hernando-Pradíes, Silvia Santamaría, David Herrera, Jose A. Garcia-Sanz, Mariano Sanz

**Affiliations:** 1ETEP (Etiology and Therapy of Periodontal and Peri-implant Diseases) Research Group, School of Dentistry, University Complutense, 28040 Madrid, Spain; nereasanchez@ucm.es (N.S.); fabiovig@ucm.es (F.V.); isanzmartin@gmail.com (I.S.-M.); coca125@hotmail.com (A.C.); javilds@hotmail.com (J.N.); javier.sanz.esporrin@ucm.es (J.S.-E.); irenhe02@ucm.es (I.H.-P.); davidher@ucm.es (D.H.); 2Department of Anatomy and Embryology, Faculty of Medicine, University Complutense, 28040 Madrid, Spain; emaldonado@ucm.es; 3Margarita Salas Center for Biological Research (CIB-CSIC), 28040 Madrid, Spain; silviasgm2@gmail.com (S.S.); jasanz@cib.csic.es (J.A.G.-S.)

**Keywords:** cell therapy, mesenchymal stem cells, mesenchymal stromal cells, gingiva-derived stem cells, soft tissue regeneration, gingival recession, root coverage, gingival recession

## Abstract

(1) Background: To investigate the effect of a xenogeneic collagen matrix (CMX) seeded with autologous gingiva-derived mesenchymal cells (GMSCs) when combined with a coronally advanced flap (CAF) in the treatment of localized gingival recession type 1 (RT1). (2) Methods: Dehiscence-type defects were created in seven dogs. GMSCs were isolated, transfected with a vector carrying green fluorescent protein (GFP) and expanded. Once chronified, the defects were randomly treated with (1) CAF plus the combination of CMX and GFP^+^ GMSCs, (2) CAF plus CMX with autologous fibroblasts, (3) CAF plus CMX and (4) CAF alone. Histological and clinical outcomes at 2- and 6-week healing periods were analyzed and compared among groups. (3) Results: Histologically, the addition of autologous cells to the CMX resulted in reduced inflammation and a variable degree of new cementum/bone formation. CMX plus GMSCs resulted in greater mean recession reduction (1.42; SD = 1.88 mm) and percentage of teeth with recession reduction of ≥2 mm (57%) when compared to the other groups, although these differences were not statistically significant. (4) Conclusions: The histometric and clinical results indicated a positive trend favouring the combination of CMX and GMSCs with the CAF when compared to the groups without cells, although these differences were not statistically significant.

## 1. Introduction

The current gold-standard treatment for root coverage in localized gingival recessions is the coronally advanced flap (CAF) with the addition of a connective tissue graft, mainly in teeth with a thin gingival phenotype. This combination has shown greater efficacy in terms of complete root coverage and long-term stability of the gingival margin when compared to the CAF alone [1]. The harvesting of a connective tissue graft, however, often results in increased patient morbidity and may have limitations due to the amount of donor tissue needed, mainly in the treatment of multiple adjacent localized gingival recessions [2]. Furthermore, the graft integration may affect the patient’s aesthetic appearance due to impaired blending and color matching with adjacent tissues [3].

Due to these limitations, xenogeneic and allogeneic soft tissue substitutes have been developed and evaluated both in preclinical and clinical studies. Histologically, porcine-derived collagen matrixes (CMX), when combined with the CAF, have shown good histological integration with adjacent tissues and lack of significant inflammatory reaction [4,5]. Clinically, evidence from clinical trials has shown comparable results when comparing the CAF/CMX combination with the CAF alone in the treatment of single [6,7] and multiple gingival recession lesions [8]. Nevertheless, when compared to the CAF combined with a connective tissue graft, inferior results have been reported in terms of mean recession reduction (RecRed) and percentage of root coverage [9,10,11]. To improve the outcomes of CMX, tissue-engineering strategies have been proposed, mainly by combining the soft tissue integration and scaffolding effect of the CMX with the addition of growth factors or precursor cells [12,13,14].

Mesenchymal stromal cells (MSCs) are multipotent cells with proliferative and paracrine capabilities, able to differentiate into multiple cell lineages [15,16] and to modulate the host response [17]. MSCs isolated from different tissues, such as bone marrow, dental pulp or periodontal ligament, have demonstrated regenerative potential in the treatment of osseous and periodontal defects, both in pre-clinical experimental models [18,19,20] and in clinical studies [21,22]. In soft tissue regeneration, a combination of the CAF with allogeneic umbilical cord MSCs embedded in polylactic-co-glycolic acid (PLGA) membranes were compared to the CAF plus PLGA membrane in a randomized controlled clinical trial treating multiple recession defects, reporting significantly superior outcomes in mean root coverage in the test group [14]. Although this single clinical study has demonstrated a beneficial effect of the MSCs therapy, evidence on the regenerative effect of cell therapies in soft tissue reconstruction of localized gingival recessions is lacking, since there are no preclinical in vivo studies reporting the histological outcomes for the use of MSCs.

It was, therefore, the purpose of this preclinical in vivo investigation to evaluate the soft tissue regenerative potential of gingiva-derived MSCs (GMSCs) seeded in CMX when combined with the CAF procedure in the treatment of single recession-type defects as compared to the CAF plus a CMX seeded with fibroblasts, the CAF plus CMX alone and the CAF alone.

## 2. Results

### 2.1. Cell Expansion and Differentiation

Autologous canine GMSCs were cultured and expanded as previously described for human GMSCs [23]. Expression of the markers defining the MSCs phenotype could not be analyzed since the monoclonal antibodies recognize human antigens but not the corresponding canine antigens. However, the cells could be differentiated into adipocytes, osteoblasts and chondroblasts (Figure 1). GMSCs and fibroblasts (FBs) exhibited a fibroblastic shape and colony-forming ability without significant morphological differences between them. After expansion, 2 × 10^7^ GMSCs were seeded onto the matrix in the test group, and a similar number of expanded fibroblasts were seeded onto the matrix using the same conditions in the control group (Figure 1E).

### 2.2. Histological Observations

#### 2.2.1. Two Weeks of Healing

A mild inflammatory infiltrate was observed in the supracrestal connective tissue portion in the CAF, CMX+GMSCs and CMX+FBs treatment groups (Figure 2e,f,j–m). This inflammatory infiltrate was more intense in the CMX group, especially in the area lateral to the junctional epithelium and within the CMX (Figure 2h,i). The CMX was well integrated in the subjacent connective tissue, showing areas of degradation in most of the samples (Figure 2h–m). New blood vessels and fibroblasts were identified within the matrix in the three treatment groups using this scaffold (Figure 2i,k,m). No signs of foreign body response or tissue necrosis were identified in any section, although signs of root resorption could be identified; this often occurred adjacent to areas where the cementum was scaled during the surgical intervention (Figure 2g). Areas of new connective tissue adhesion, cementum and bone were identified in all treatment groups, although areas of an interface with a long junctional epithelium without traces of new cementum were also found in all treatment groups.

#### 2.2.2. Six Weeks of Healing

The four treatment groups demonstrated a healthy supracrestal connective tissue with minimal inflammatory infiltrate lateral to the junctional epithelium. Remnants of the collagen matrix were still identifiable in specimens from all the groups using this scaffold (Figure 3b,d–f). In the CMX+GMSCs group, a unique group of dark stained cells were identified, suggesting the influence of green fluorescent protein (GFP)-positive GMSCs on the formation of new periodontal tissues (Figure 3h,i), unlike the control sample, in which no dark staining could be found after immunohistochemistry for GFP (Figure 3g).

At 6 weeks, different healing patterns could be identified in different sections irrespective of the treatment group (Figure 4). In some sections there was a pattern of healing by long junctional epithelium without traces of new cementum, while in others there was a clear regenerative pattern with clear identification of new cementum, new connective tissue attachment and new bone formation (Figure 4a,e). There was even a third pattern characterized by new bone formation in the absence of new cementum formation (Figure 4b,g). It may be suggested, therefore, that root coverage surgical procedures using a coronally advanced flap may induce a certain degree of periodontal regeneration, although frequently combined with a reparative process through a long junctional epithelium. These healing patterns were not modified with the addition of a collagen matrix with or without the addition of autologous cells. The outcome of these healing patterns was not predictable and did not depend on the treatment group. 

### 2.3. Histometric Analysis 

Out of a total of 56 specimens, 49 were subjected to a histometric analysis (Table 1). Two samples from the CMX+GMSCs group were excluded (due to suture dehiscence in a 2-week sample and due to tissue damage during the histological processing in a 6-week specimen) and four could only be partially studied (two from the CAF and two from the CMX+FBs group), also due to histological processing errors. The following landmarks were used for the histometric measurements (Figure 3a):M: The most coronal marginal mucosa;aJE: The most apical extent of the junctional epithelium;cC: The most coronal extent of the cementum;cN: The most apical portion of the coronal notch at the gingival margin level;aN: The most apical portion of the apical notch;Bc: The most coronal level of the Bc.

**Table 1 ijms-23-03248-t001:** Histometric measurements at the 2- and 6-week healing times.

Outcome Variable	Healing Period	CAF (*n* = 14)	CAF+CMX (*n* = 14)	CAF+CMX+GMSCs (*n* = 13)	CAF+CMX+FBs (*n* = 14)	*p*-Value *
M–aJE in mm, mean (SD)	2 wk	4.97 (1.45)	4.37 (1.55)	5.32 (1.49)	4.87 (1.90)	0.766 ^‡^
	6 wk	3.51 (2.09)	3.24 (1.47)	3.91 (2.29)	4.57 (1.19)	0.593 ^‡^
cN–M in mm, mean (SD)	2 wk	2.54 (0.92) *	0.61 (1.09) *	1.31 (0.54)	1.09 (0.72) *	0.003 ^‡^
	6 wk	0.93 (0.82)	1.02 (0.99)	0.88 (1.94)	1.22 (0.56)	0.950 ^‡^
aN–aJE in mm, mean (SD)	2 wk	1.62 (3.44)	0.61 (1.20)	−0.031 (0.062)	0.96 (1.25)	0.177 ^†^
	6 wk	1.52 (1.59)	1.69 (0.99)	1.46 (1.04)	1.30 (2.13)	0.498 ^‡^
aN–cC in mm, mean (SD)	2 wk	0.08 (0.21)	0.59 (1.21)	0.00 (0.00)	1.09 (0.72)	0.701 ^†^
	6 wk	1.43 (1.58)	1.58 (1.07)	1.35 (1.17)	0.40 (0.65)	0.252 ^‡^
aN–Bc in mm, mean (SD)	2 wk	−0.48 (0.61)	−0.41 (0.84)	−0.50 (0.75)	−0.034 (1.11)	0.825 ^†^
	6 wk	−0.14 (0.62)	−0.018 (0.99)	−0.207 (0.50)	0.03 (0.74)	0.948 ^‡^

CAF, coronally advanced flap; CAF+CMX, coronally advanced flap + collagen matrix; CAF+CMX+GMSCs, coronally advanced flap + collagen matrix + gingiva-derived mesenchymal stem cells; CAF+CMX+FBs, coronally advanced flap + collagen matrix + fibroblasts; SD, standard deviation; M, most coronal marginal mucosa; aJE, most apical extent of the junctional epithelium; cN, coronal notch; cC: most coronal extent of the cementum; aN, apical notch; Bc, bone crest; M–aJE, length of the junctional epithelium; cN–M: histological “root coverage” or soft tissue gain above the coronal notch; aN–aJE: length of connective tissue attachment; aN–cC, length of newly formed cementum; aN–Bc, length of new bone. * The CAF treatment group showed a greater mean cN–M dimension than the CAF+CMX and CAF+CMX+FBs groups (*p* < 0.05). ^†^ Kruskal–Wallis tests were used for continuous variables with non-normal distributions. ^‡^ One-way ANOVA tests was used for continuous variables with normal distributions. Statistically significant differences were considered for *p* ≤ 0.05.

#### 2.3.1. Epithelial Length (M–aJE)

Although no significant differences among groups were found, the CMX group was slightly better in terms of mean epithelial length, with lower dimensions for M–aJE at 2 (4.37 mm; SD = 1.55) and 6 weeks (3.24 mm; SD = 1.47). However, the CMX+GMSCs group (5.32 mm; SD = 1.49) at 2 weeks and the CMX+FBs group (4.57 mm; SD = 1.19) at 6 weeks exhibited poorer outcomes than the other treatment groups (*p* > 0.05) (Table 1).

#### 2.3.2. Histological Root Coverage (cN–M)

The CAF group exhibited larger dimensions of histological root coverage (2.54; SD = 0.92) than the CMX (0.61; SD = 1.09) and the CMX+FBs (1.09; SD = 0.72) groups at 2 weeks, with statistically significant differences between them (Table 1).

#### 2.3.3. Length of the Connective Tissue Attachment (aN–aJe)

Despite the lack of statistically significant differences between the groups, at 2 weeks, the CMX group showed better results than the other three groups, since it achieved the greatest dimensions of connective tissue attachment (1.62; SD = 3.44). At 6 weeks, the four groups exhibited similar results, ranging from 1.30 to 1.69 mm (*p* > 0.05) (Table 1).

#### 2.3.4. Length of Newly Formed Cementum (aN–cC)

At 2 weeks, the CMX+FBs group showed a higher dimension of new cementum formation, approximately 1 mm (1.09; SD = 0.72). However, at 6 weeks, this group exhibited inferior results for this outcome variable (0.40; SD = 0.65) in comparison with the other groups demonstrating cementum gains of at least 1 mm (*p* > 0.05) (Table 1).

#### 2.3.5. Dimensions of New Bone (aN–Bc)

The only treatment group that showed mean bone gains at 2 weeks was the CMX+FBs group (*p* > 0.05). At 6 weeks, all groups demonstrated similar bone gain dimensions (Table 1). 

### 2.4. Clinical Observations

The wound healing after the surgical interventions occurred uneventfully in all the animals, except in one tooth allocated to the CMX+GMSCs group at 2 weeks that showed a soft tissue dehiscence; therefore, this site was excluded from the analysis. 

The four treatment groups were homogeneous at baseline in terms of the depth of the recession (Rec). Table 2 depicts the results in the clinical outcome variables over time. All treatment groups demonstrated statistically significant reductions in recession (RecRed, primary outcome variable) (Appendix A Table A1). The CAF alone group showed the highest RecRed between baseline and 2 weeks (2.85 ± 0.89 mm), while the CMX+GMSCs group demonstrated the highest RecRed between baseline and 6 weeks (1.42 ± 1.88 mm); however, these differences were not statistically significant between groups (Table 2). Similarly, in terms of complete root coverage, the CAF alone showed the highest percentage between baseline and 2 weeks, but between baseline and 6 weeks, both the CMX+GMSCs and CMX+FBs groups demonstrated the highest percentages (57.1%), though these differences were not statistically significant (Table 2). Analogously, the percentage of sites with RecRed ≥2 mm between baseline and 6 weeks was higher for the CMX+GMSCs group (about 57%) compared with the other treatment groups, though differences, again, were not statistically significant (Appendix B Table A2).

Appendix C Table A3 depicts the clinical results by tooth type, showing that the lower canines were the teeth with the least RecRed, while the upper canines were the teeth with the highest RecRed and percentage of complete root coverage between baseline and 6 months.

## 3. Discussion

The aim of this pilot preclinical investigation was to analyze the added benefit of adding a xenogeneic CMX seeded with GMSCs as compared to adding mature fibroblasts or using a CMX without cells or when only using the CAF in the treatment of localized gingival recession defects in terms of histological and clinical outcomes. 

Regarding histological outcomes, the collagen matrix at 2 weeks showed good integration within the connective tissue bed. These histological outcomes are consistent with those described at 1 week in a preclinical in vivo study by Vignoletti et al. using the same xenogeneic matrix, without the addition of any cells, in combination with the CAF for the treatment of localized gingival recession defects [5]. At this time point, the seeding of the xenogeneic CMX and autologous cells (fibroblasts or GMSCs) did not result in increased inflammation or presence of areas of necrosis or foreign tissue reaction. In fact, when sections from the CMX alone group were compared to those belonging to the CMX+GMSCs and CMX+FBs groups, the CMX group showed a denser inflammatory infiltrate. This finding could be related to the immunosuppressive properties of MSCs [24,25]. Similarly, some publications have attributed immunomodulatory potential to fibroblasts due to their effect of inhibiting the proliferation of peripheral blood mononuclear cells and the stimulation of the synthesis of interferon-gamma [26]. Therefore, we may suggest that the use of GMSCs or fibroblasts seeded in collagen matrixes could be a valuable therapeutic tool for improving wound healing and reducing inflammation-related postoperative complications associated with mucogingival surgery, such as discomfort, pain and swelling. 

All treatment groups displayed different degrees of newly formed cementum, bone and a connective tissue attachment, but there were also sections depicting a long junctional epithelium with no cementum and bone formation. Our results are equivalent to those reported by Vignoletti et al. [5], since, at 4 weeks, both treatment groups demonstrated a similar healing pattern in terms of new cementum formation. However, when a longer follow-up was evaluated (3 months), they reported increased dimensions of new cementum in the group using the CMX. The short follow-up at which to properly evaluate the histological outcomes may, therefore, be a limitation of this preclinical investigation. 

In terms of the clinical results, although no statistically significant differences were found among groups, the CMX+GMSCs group showed a trend towards the greatest RecRed values (1.42; SD = 1.88 mm), a higher percentage of complete root coverage (57.,1%) and a higher percentage of sites with ≥2 mm RecRed (57%) between baseline and 6 weeks (Table 2). However, the CAF and CMX+FBs groups attained higher percentages of root coverage, although these differences were not statistically significant (*p* > 0.05). Although there are no published preclinical investigations evaluating the added clinical effect of cell therapies using MSCs in root coverage procedures, several clinical trials have evaluated different cell therapy strategies for the treatment of recession-type defects [14,27,28,29,30,31,32]. Most of these studies are randomized controlled clinical trials or case series, in which Miller I and II defects [33] (currently RT1 defects) [34] were treated by a CAF and a subepithelial acellular dermal matrix or a xenogeneic CMX with embedded cells, frequently fibroblasts, in comparison to the CAF with either a connective tissue graft or the CMX alone [14,27,28,29,31,32]. In only two of these randomized controlled clinical trials, the effect of MSCs was evaluated in root coverage procedures [14,32]. In a first study, 6 weeks after the intervention, the test group (CAF+PLGA+allogeneic umbilical cord MSCs) exhibited a slightly lower RecRed (1.76; SD = 0.60 mm) than the control group (CAF+connective tissue graft) (2.10; SD = 0.67 mm) in the treatment of multiple recession defects [32]. In a later publication, the same test treatment exhibited a significantly higher mean RecRed than the control group using the scaffold without the cells (1.57; SD = 0.89 vs. 1.24; SD = 0.47 mm). 

Within the limitations of this preclinical investigation, the use of a CAF with a subepithelial CMX with embedded GMSCs resulted in an uneventful healing, a good integration within the connective tissue and a lower inflammatory response when compared with the use of CAF and the CMX alone. The healing of this cell therapy construct was characterized by different repair phenomena, since, in several samples, the reconstruction of the periodontal attachment with new cementum, new bone and new connective tissue attachment was identified, whilst, in others, there was a reparative long junctional epithelium. Although the histometric variables, as well as the clinical outcomes, showed a positive trend favoring the CMX+GMSCs treatment after 6 weeks, the results were similar between the groups, with no statistically significant differences. According to the results of the present study, for the time being, the use of MSCs is not cost-efficient. Further experimental in vivo studies should be carried out with larger samples and longer healing times to properly evaluate the real biological potential of this cell therapy.

## 4. Materials and Methods

### 4.1. Study Design and Randomization

This study was designed as a preclinical split-mouth randomized controlled pilot study comparing four root coverage surgical protocols: (1) CAF plus a xenogeneic CMX (Mucograft^®^, Geistlich Pharma AG, Wolhusen, Switzerland) seeded with autologous GMSCs (test group; CMX+GMSCs), (2) the CAF plus a CMX seeded with autologous fibroblasts (control group 1; CMX+FBs), (3) CAF plus a CMX (control 2; CMX) and (4) CAF alone (control 3; CAF). Each animal provided eight study sites, four in the maxilla and four in the mandible (upper second incisors, upper canines, lower canines and the mesial roots of lower first molars) and two healing times (2 and 6 weeks). Each site was randomly assigned to one intervention and healing time using a computer-generated randomization list stratified by hemi-mandible/maxilla, tooth and healing time (IBM SPSS Statistics^®^ v20. JM. Domenech). Allocation to the tested interventions was concealed by using sealed envelopes, which were opened during the surgical procedure once the flaps were raised.

### 4.2. Experimental Sample

Seven 12-month-old beagle dogs (four males, three females), with an approximate weight of 9–12 kg, were used in full compliance with the ARRIVE guidelines and the Spanish and European regulations concerning the use and care of research animals. All experimental animals were acquired from Marshall BioResources (North Rose, NY, USA) and housed in purpose-designed cages for large experimental animals (Experimental Surgical Center, Gómez Ulla Hospital, Madrid, Spain).

The Ethical Committee of the Gómez Ulla Central Military Hospital (Madrid, Spain) approved the study protocol. The dogs were maintained in a group kennel with outdoor and indoor areas, with a controlled temperature of 18 ± 2 °C and natural light and air renewal, and were monitored daily by a veterinarian accredited in laboratory animal science. The animals were fed using a granulated dog food, previously wetted in water, with individual bowls and free supply of water, and subjected to plaque control twice a week after the root coverage intervention using gauzes soaked with chlorhexidine 0.12% (Perio-Aid^®^, Dentaid, Cerdanyola, Spain). After an adaptation/quarantine period of 3 weeks, the experimental segment of the study took place.

### 4.3. Treatment Protocols

All surgical interventions were performed in an animal operating theatre under aseptic conditions. The animals were first pre-medicated with medetomidine (20 μg/kg/i.m., Domtor, Esteve, Barcelona, Spain) and morphine for pain control (0.4 mg/kg/i.m., Morfina Braun 2%, B. Braun Medical, Barcelona, Spain). Then, general anesthesia was induced with propofol (3–5 mg/kg/i.v., Propovet^®^, Abbott Laboratories, Kent, UK) and maintained with a concentration of 2.5–4% isoflurane (Isoba-vet^®^, Schering-Plough, Madrid, Spain). During anaesthesia, the animals were cared for by a veterinarian doctor (B or C category), who continuously monitored the animals with electrocardiography, capnography, pulsi-oximetry and non-invasive blood pressure measurement. The surgical areas were locally anaesthetized (0.4% articaine/epinephrine 1:100000, Inibsa, Barcelona, Spain). 

Prophylactic cefazolin (20 mg/kg/i.v., Kurgan, Normon, Madrid, Spain) and cefovezin (8 mg/kg/s.i.d./s.c., Convenia, Zoetis, Madrid, Spain) were administered intraoperatively.

At the end of the intervention, atipamezole (50 mg/kg/i.m., Esteve, Barcelona, Spain) was administered to revert the effects of medetomidine. Post-operative pain was controlled by administration of morphine (0.2 mg/kg/i.m./6 h, Morfina Braun 2%, B. Braun Medical, Barcelona, Spain) and meloxicam as an anti-inflammatory and analgesic treatment (0.2 mg/kg/i.m./SID, Metacam, Boehringer Ingelheim, Barcelona, Spain) for five days.

#### 4.3.1. Phase 1: Tooth Extraction and Cell Isolation

After removing biofilm and calculus deposits with ultrasonic scalers, the distal roots of lower first molars and fourth premolars were extracted once hemisected using forceps and root elevators. The exposed pulp was then sealed with calcium hydroxide (Dycal^®^, Densply, York, PA, USA) and a glass ionomer cement restoration (Ketac^®^ Cem, 3M ESPE, Berkshire, UK).

Gingival connective tissue samples (5 × 5 × 2 mm) were obtained from the palate using a circular scalpel (Omnia, Fidenza, Italy) and de-epithelialized. Then, dermis biopsies (5 × 5 × 2 mm) were taken from the back skin of the animals.

Gingival tissue specimens were processed, cultured and differentiated to GMSCs using a protocol already described by our research team in a previous publication [23]. In brief, after tissue digestion, the connective tissue was centrifuged and the pellet resuspended in DMEM:F12 supplemented with 10% FCS, 100 U/mL penicillin, 100 μg streptomycin and 2 mM L-glutamine (Sigma-Aldrich, St. Louis, MO, USA). 

At the first passage, GMSCs were transfected with an expression vector carrying the GFP cDNA under the control of the eukaryotic elongation factor alpha promoter, plasmid pEF1 prom-eGFP, using standard protocols. GFP^+^ cells were selected by cell sorting and subsequently expanded. The resulting autologous canine GMSCs (2 × 10^7^ at the third passage) and autologous fibroblasts (2 × 10^7^) were incubated for 30 min (37°, 5% CO_2_ and 95% humidity) into 15 × 10 matrixes. The processing and expansion protocol of the dermal fibroblasts was identical to the one described for GMSCs. 

To demonstrate their stem cell phenotype, GMSCs were differentiated to osteogenic, adipogenic and chondrogenic lineages using specific media [23]. 

#### 4.3.2. Phase 2: Experimental Dehiscence-Type Defects

Dehiscence-type defects were experimentally developed according to the protocol described by Vignoletti et al. [5]. In brief, after eliminating the redundant marginal soft tissue while maintaining a band of keratinized tissue (approximately 3 mm), two horizontal incisions (3 mm in length) were performed at both aspects of the tooth followed by two vertical releasing incisions (Figure 5). After raising a full-thickness flap, 5 mm of the buccal bone/cementum, apical to the cemento-enamel junction (CEJ), was removed. The flaps were then apically repositioned and sutured (Vicryl 5-0, Ethicon, Somerville, MA, USA). 

#### 4.3.3. Phase 3. Root Coverage Surgeries

Four weeks later, once the combined hard and soft tissue defects were chronified, the coronally advanced flap surgical design [35] was used to treat all the localized gingival recessions. 

GMSCs and FBs were detached from the culture flask and the number of living cells counted with an automated cell counter (TTC model CASY^®^, 150 μm capillary, Roche Diagnostics, Basel, Switzerland) [36]. Then, 2 × 10^7^ GMSCs were placed onto a matrix, incubated at 37 °C for 30 min and allowed to seed. Afterwards, the matrix was transferred onto another well, and media was added to cover the matrix and cells. The same protocol was followed for seeding fibroblasts into the matrix. 

In the target teeth, two horizontal incisions at the mesial and distal aspects of the CEJ were followed by two beveled vertical incisions beyond the mucogingival junction. Once the flaps were elevated following a split–full–split thickness approach, a coronal notch was prepared on the root surface at the gingival margin and a second notch at the bone crest. Then, the exposed root surface was debrided, the anatomical papillae were de-epithelialized and the CMX was adapted and sutured with t-mattress resorbable sutures (Vicryl 5-0, Ethicon, Somerville, MA, USA). Then, the flap was coronally positioned 1 mm above the CEJ using sling sutures, supplemented with interrupted sutures (Vicryl 5-0, Ethicon, Somerville, MA, USA) (Figure 5). 

All the surgical interventions were carried out by experienced and trained surgeons (I.S, F.B, J.N) blinded to the group allocation.

#### 4.3.4. Phase 4: Euthanasia

Six weeks after the last root coverage intervention, the experimental animals were first sedated with medetomidine (30 μg/kg/i.m., Esteve, Barcelona, Spain) and then euthanized with an intravenous overdose of sodium pentobarbital (40–60 mg/kg/i.v., Dolethal, Vetoquinol, France). Before fixation, the upper and lower jaws were dissected, and the tested teeth with their intact soft tissues were individually separated and retrieved using a band saw. Subsequently, the specimens were fixed in buffered 10% formaldehyde solution (Exakt, Norderstedt, Germany).

### 4.4. Histological Processing

The neutral-buffered formalin specimens were sectioned into 3–8 mm wide bucco-lingual blocks using disks (911 HH Ø 22 mm 0.1 mm L disks, Komet, Besigheim, Germany). These blocks were then decalcified for 8–9 months in Osteosoft^®^ (Sigma-Aldrich, St. Louis, MO, USA), embedded in paraffin, cut in 5 mm thickness sections, and stained with haematoxylin and eosin (H&E), Masson´s trichrome acc. (McFarlane, Martilengo, Italy), Movat’s pentachrome (ScyTek Laboratories, UT, USA) and Picrosirius red (Abcam, Cambridge, UK). Selected sections were also processed for immunohistochemistry, using recombinant anti-GFP antibody (EPR14104) (ab183734) at 1:250 concentration (Abcam, Cambridge, UK). 

### 4.5. Histological and Histometric Analysis

Two sections from the central aspects of each tooth were chosen for the histological analysis. All the histometric measurements were carried out by a single-blinded calibrated examiner (NS (ICC = 0.67–1.00; 95% CI)) using a Nikon Eclipse Ti microscope (Nikon, Heidelberg, Germany) equipped with the software Nis Elements BR (Nikon DS-Ri1, Amstelveen, The Netherlands). 

### 4.6. Clinical Evaluation

A trained and calibrated examiner (NS), blinded to the treatment allocation, recorded the clinical outcomes. The calibration exercise for the clinical measurements provided a high degree of reproducibility with intraclass correlation coefficients (ICC) = 0.80–0.85 (95% CI).

Gingival recession (CEJ–gingival margin) was recorded before the root coverage procedure and at the time of sacrifice with a NC15 periodontal probe (PCP-UNC 15, Hu-Friedy, Chicago, IL, USA) and rounded to the nearest millimeter (mm). The change in gingival recession (RecRed) was measured between baseline and 2 and 6 weeks and the percentage of root coverage was calculated.

### 4.7. Statistical Analysis

Means and standard deviations for histological and clinical parameters were determined for each treatment group and healing period, the treated tooth being the unit of analysis (*n* = 56).

For intergroup comparisons, at each healing time and between baseline and 2 or 6 weeks, one-way ANOVA or Kruskal–Wallis tests were utilized for quantitative variables. For categorical variables, the chi-squared test was employed. Data were analyzed using SPSS 21.0 software (Lead Technologies Inc., Charlote, NC, USA), with *p* ≤ 0.05. 

## Figures and Tables

**Figure 1 ijms-23-03248-f001:**
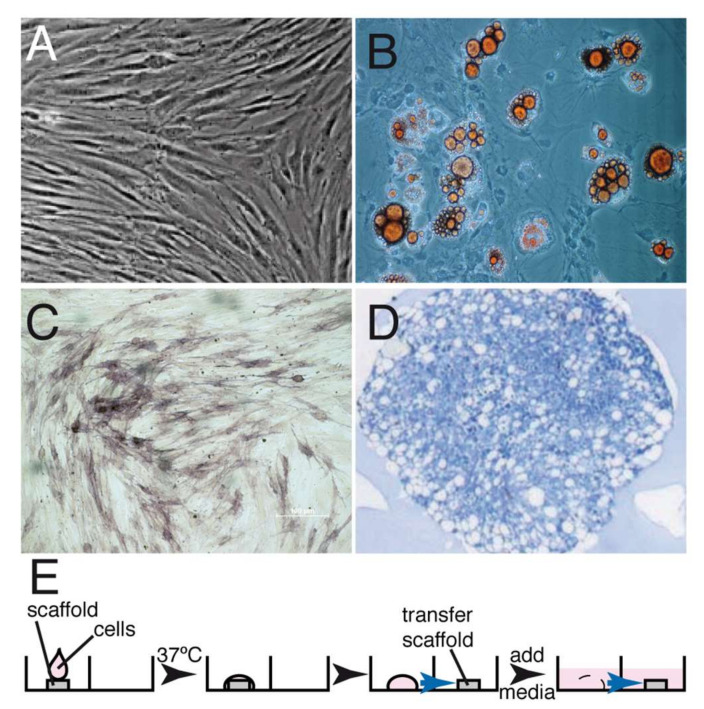
GMSC isolation, characterization and seeding into the matrix. GMSCs were isolated as described in Materials and Methods and expanded in tissue culture. (**A**) Phase-contrast micrography showing the exponentially growing GMSCs. These cells were then differentiated into fat cells (**B**), osteocytes (**C**) and chondrocytes (**D**). (**E**) Protocol describing the seeding of the undifferentiated GMSCs on the matrix; 2 × 10^7^ cells were dropped onto a matrix, incubated at 37 °C for 30 min, allowed to seed and then the matrix was transferred onto another well and media was added to cover the matrix and cells. Under the conditions used, between 50–60% of the cells were attached to the matrix.

**Figure 2 ijms-23-03248-f002:**
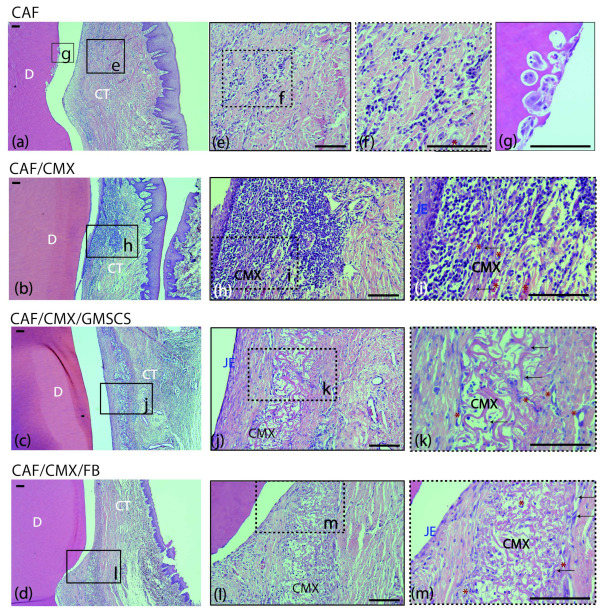
Cross-sections of the buccal dento-gingival area after 2 weeks of healing. Haematoxylin and eosin (HE) stain. (**a**–**d**) Photomicrographs of the supracrestal periodontal tissues in the four groups. (**e**,**f**) Mild acute inflammatory infiltrate in the connective tissue in the CAF group. (**g**) Image showing root resorption with presence of osteoclasts in Howship’s lacunae in the same specimen from the CAF group. (**h**,**i**) Inflammatory infiltrate in the connective tissue surrounding the matrix in the CAF+CMX group. Note the high density of leucocytes and blood channels. (**j**,**k**) Collagen matrix within the connective tissue with mild inflammatory infiltrate and few fibroblasts in the CAF+CMX+GMSCs group and in the CAF+CMX+FBs group (**l**,**m**). Please note the lower density of leucocytes in the GMSCs and FBs groups in comparison with the CMX alone group. CAF, coronally advanced flap; CAF+CMX, coronally advanced flap+collagen matrix; CAF+CMX+GMSCs, coronally advanced flap+collagen matrix+gingiva-derived mesenchymal stem cells; CAF+CMX+FB, coronally advanced flap+collagen matrix+fibroblasts; SD, standard deviation; D, dentin; CT, connective tissue; CMX, collagen matrix; JE, junctional epithelium; black arrows, fibroblasts; red asterisks, blood channels. Scale bar = 100 μm.

**Figure 3 ijms-23-03248-f003:**
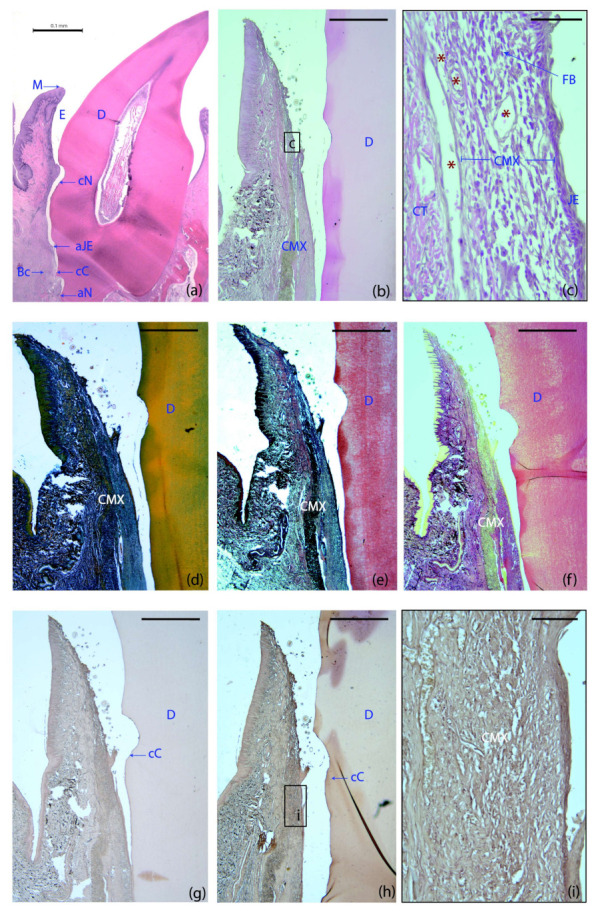
Histological landmarks and histology slides of a tooth allocated to the GMSCs group at six weeks. (**a**) Landmarks used for histometric analysis; haematoxylin and eosin (HE) stain. (**b**) Cross-section of the buccal dento-gingival region; HE stain. (**c**) High-magnification images of the gingival area containing the matrix. (**d**) Masson’s trichrome stain. (**e**) Movat’s pentachrome stain. (**f**) Picrosirius red stain. (**g**,**h**,**i**) Immunohistochemistry for green fluorescent protein (GFP): control (**g**) and GFP-labeled samples (**h**). (**i**) High magnification image of the GFP-labeled sample. Note the different color intensity among the control and the GFP-labeled groups, demonstrating the influence of GFP^+^ cells on the formation of new periodontal tissues. M, the most coronal marginal mucosa; aJE, the most apical extent of the junctional epithelium; cC, the most coronal extent of the cementum; cN, the most apical portion of the coronal notch at the level of the gingival margin; aN, the most apical portion of the apical notch; Bc, the most coronal level of the bone crest; E; area where the enamel was before decalcification; D, dentin; CMX, collagen matrix; JE, junctional epithelium; CT, connective; FBs, fibroblasts; red asterisks, blood vessels. Scale bar = 0.1 mm (**a**); 1000 μm (**b**–**g**); 50 μm (**c**); 100 μm (**i**).

**Figure 4 ijms-23-03248-f004:**
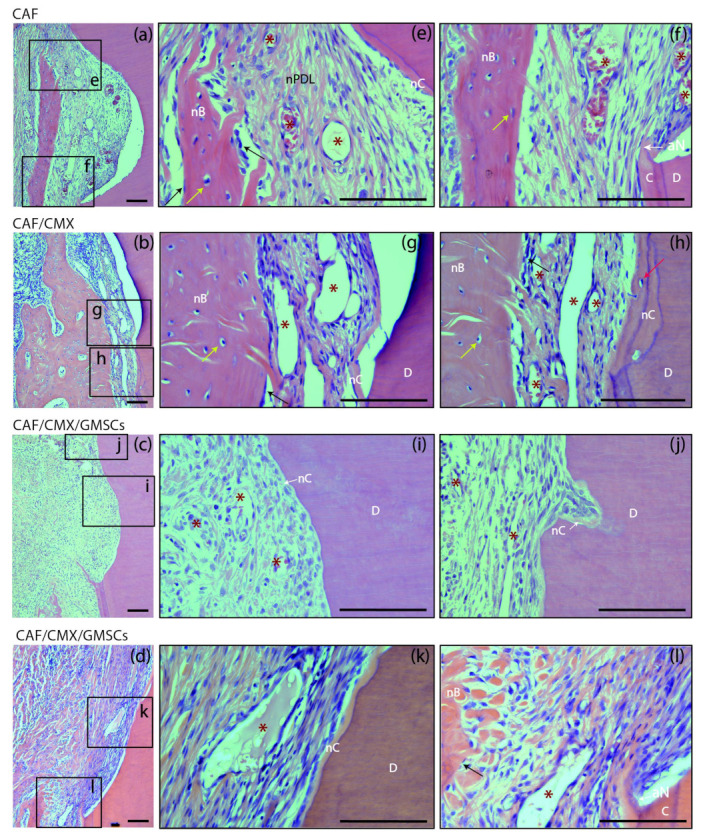
Bucco-lingual histological sections depicting the periodontal tissues at the level of the apical notch at 6 weeks; haematoxylin and eosin (HE) stain. (**a**–**d**) Images of the apical notch area in specimens from the four groups. (**e**,**f**) High-magnification images showing the newly formed cementum, bone and periodontal ligament in a specimen from the CAF group. (**g**,**h**) Images from a sample from the CAF+CMX group in which bone growth has extensively occurred despite the limited dimension of new cementum. (**i**) Newly formed connective tissue adhesion and cementum in a specimen from the GMSCs group. (**j**) New cementum formation in a resorption lacuna. (**k**,**l**) High-magnification pictures showing the histological attachment level gain associated with the newly formed cementum and the immature bone coronal to the apical aspect of the apical notch. CAF, coronally advanced flap; CAF+CMX, coronally advanced flap + collagen matrix; CAF+CMX+GMSCs, coronally advanced flap + collagen matrix + gingiva-derived mesenchymal stem cells; CAF+CMX+FBs, coronally advanced flap + collagen matrix + fibroblasts; SD, standard deviation; nB, new bone; nC, new cementum; nPDL, new periodontal ligament; C, cementum; D, dentin; aN, apical aspect of the apical notch; asterisks, blood vessels; black arrows, osteoblasts; yellow arrows, osteocytes; red arrow, cementoblast. Scale Bar = 100 μm.

**Figure 5 ijms-23-03248-f005:**
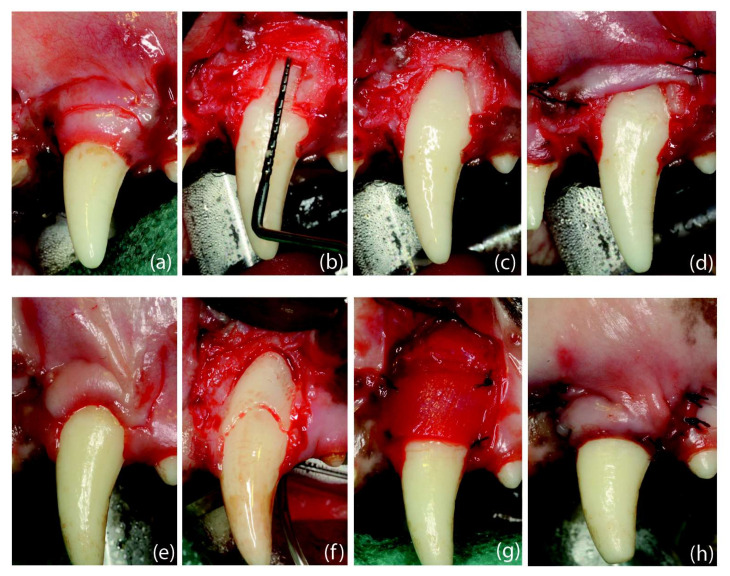
Surgical interventions: (**a**) dehiscence-type defect creation; (**b**) root coverage procedure. Defect creation: (**a**) Scalloped incision, preserving 3 mm of keratinized tissue, followed by two horizontal incisions (2 mm) and two vertical releasing incisions extending 5 mm in the alveolar mucosa. (**b**) Removal of the buccal bone until reaching a vertical distance from the cemento-enamel junction to the mid-buccal bone of 5 mm. (**c**) Bone removal until showing the mesio-distal dimension of the root. (**d**) Flap sutured in an apical position. Root Coverage Surgery: (**e**) Design of the coronally advanced flap (4 weeks after defect creation) by performing two horizontal beveled incisions (at the level of the buccal cemento-enamel junction) at the mesial and distal aspects of the tooth with the recession defect followed by two vertical, slightly divergent releasing incisions. (**f**) Coronal notch performed at the level of the gingival margin and apical notch at the level of the bone crest, after flap-raising with a split–full–split approach; de-epithelialized papillae. (**g**) Suture of the collagen matrix with simple stitches. (**h**) Flap sutured in a coronal position.

**Table 2 ijms-23-03248-t002:** Clinical variables at baseline and 2 and 6 weeks after the root coverage procedure.

Variable	CAF (*n* = 14)	CAF+CMX (*n* = 14)	CAF+CMX+GMSCs (*n* = 13)	CAF+CMX+FBs (*n* = 14)	*p*-Value
**Recession length in mm** **mean (SD)**					
Baseline	2.28 (0.97)	1.60 (0.88)	2.07 (1.26)	1.89 (0.81)	0.289 ^†^
2 weeks	0	0.57 (0.67)	0.25 (0.41)	0.14 (0.37)	0.095 ^†^
6 weeks	0.57 (0.60)	0.78 (0.69)	0.57 (0.93)	0.57 (0.83)	0.816 ^†^
**RecRed in mm** **mean (SD)**					
Baseline–2 weeks	2.85 (0.89)	1.07 (1.39)	1.75 (1.54)	1.85 (1.02)	0.085 ^‡^
Baseline–6 weeks	1.14 (0.37)	0.78 (0.90)	1.42 (1.88)	1.21 (0.63)	0.749 ^‡^
**Root coverage in %** **mean (SD)**					
Baseline–2 weeks	100 (0.0)	46.42 (58,50)	82.66 (28.90)	92.85 (18.89)	0.062 ^†^
Baseline–6 weeks	72.61 (28.34)	45.23 (45.86)	47.14 (94.28)	75.00 (38.18)	0.586 ^†^
**Specimens with CRC in %**					
2 weeks	100	42.9	60	85.7	0.077 *
6 weeks	42.9	28.6	57.1	57.1	0.664 *

CAF, coronally advanced flap; CMX, collagen matrix; GMSCs, gingival mesenchymal stromal cells; FBs, fibroblasts; SD, standard deviation; RecRed, recession reduction; CRC, complete root coverage. ^†^ Kruskal–Wallis tests were used for quantitative variables with non-normal distributions. ^‡^ One-way ANOVA was used for quantitative variables with normal distributions. * Chi-squared tests were used for categorical data. Statistically significant differences were considered for *p* ≤ 0.05.

## Data Availability

Not applicable.

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
