# Peer review of "Cell Therapy Based on Gingiva-Derived Mesenchymal Stem Cells Seeded in a Xenogeneic Collagen Matrix for Root Coverage of RT1 Gingival Lesions: An In Vivo Experimental Study"

_ijms, 2022, doi:10.3390/ijms23063248_

Round 1
Reviewer 1 Report
Abstract - I don't know what the authors mean by chronified but otherwise this is a clear abstract.
Results - I appreciate the strong quantification and clear images the authors have provided. I know that the soft/hard tissue interface histology is not easy.
Methods - I may have missed it, but how do the authors take the samples with cells and implant them? Do they wash the cell medium, etc? How long do they culture the cells + scaffolding constructs.
Discussion - Could have more information on what the authors speculate or hypothesize the MSCs, compared to fibroblasts, or blanks are doing to the tissue. More on immune modulation, basically. If the authors were to perform similar trials in humans (maybe they are planning to or have), what considerations are important.
Author Response
Response to Reviewer 1 Comments
We would like to thank the Reviewer 1 for his/her valuable comments.
Point 1: Abstract - I don't know what the authors mean by chronified but otherwise this is a clear abstract.
Response to Point 1: By the term “chronified” We mean that the root coverage surgeries were not performed immediately after the creation of the dehiscence-type defects but once appropriate healing had taken place (four weeks). In this way we, discard the self-regenerative capacity of the tissues and hence, we only study the regenerative efficacy of the tested cell therapies
Point 2: Methods - I may have missed it, but how do the authors take the samples with cells and implant them? Do they wash the cell medium, etc? How long do they culture the cells + scaffolding constructs.
Response to Point 2:This information was in Figure 1. We have included it also in the material and methods section (Phase 3: Root Coverage surgeries) to clarify.
“2x107GMSCs were placed onto a matrix, incubated at 37ºC for 30 min and allowed to seed. Afterwards, the matrix was transferred onto another well and media was added to cover the matrix and cells. The same protocol was followed for seeding fibroblasts into the matrix”.
All the conditions, culture media and reagents employed (for isolation, cell culture, cells detaching, etc.) are explained in detail in a previous publication from our research group (Santamaría et al. 2017). In the manuscript, the reader is referred to this publication for further information (Material and Methods-Phase 1: Tooth extraction and cell isolation).
Point 3: Discussion - Could have more information on what the authors speculate or hypothesize the MSCs, compared to fibroblasts, or blanks are doing to the tissue. More on immune modulation, basically. If the authors were to perform similar trials in humans (maybe they are planning to or have), what considerations are important.
Response to Point 3: We have added more information in the second and the last paragraph of the discussion; it is specially related to the immunomodulatory properties of both GMSCs and fibroblasts:
“This finding could be related to the immunosuppressive properties of MSCs [24,25]. Similarly, some publications have attributed immunomodulatory potential to fibroblasts by their effect of inhibiting the proliferation of peripheral blood mononuclear cells and the stimulation of the synthesis of interferon-gamma [26].Therefore, we may suggest that the use of GMSCs or fibroblasts seeded in collagen matrixes could be a valuable therapeutic tool for improving wound healing and reducing inflammation-related postoperative complications associated with mucogingival surgery, as discomfort, pain and swelling”.
For the time being, we consider that the next step is to perform further experimental studies before going for human trials (the evidence that supports improved histological and clinical outcomes for GMSCs in this study is still weak):
“Further experimental in vivo studies should be done with larger samples and longer healing times to properly evaluate the real biological potential of this cell therapy”.
Please see the attachment: modified manuscript

Reviewer 2 Report
I appreciated so much the article titled “Cell therapy based on gingiva-derived mesenchymal stem cells seeded in a xenogeneic collagen matrix for root coverage of RT1 gingival lesions: an in vivo experimental study”, developing a preclinical split-mouth randomized controlled study, comparing four root coverage surgical protocols; used seven animals with eight sites each (56 sites total) (upper second incisors, upper canines, lower canines and the mesial roots of lower first molars) and two healing times (2, 6 weeks). The study was very well developed, and the theme had great expression and importance.
After the read, I raised some concerns.
1. Abstract: “Dehiscence-type defects were created on seven dogs” and four groups “1) CAF plus the combination of CMX and GFP+ GMSCs, 2) CAF plus CMX 20 with autologous fibroblasts, 3) CAF plus CMX, 4) CAF alone” analyzed after “2 and 6 week-healing periods.”
Results - “differences weren’t statistically significant.”
The conclusion must be improved because it may cause a misunderstanding “effect of combining CMX and GMSCs with the CAF resulted in positive histological and clinical outcomes”; the values were not significant.
2. INTRODUCTION: the introduction is direct and well-written (around 50% of the articles cited within five years)
3. OBJECTIVE: “evaluate the soft tissue regenerative potential of gingiva-derived MSCs (GMSCs) seeded in CMX when combined with the CAF.”
4. M&M: there was randomization; allocation to the tested interventions was concealed by using sealed envelopes; followed the ARRIVE guidelines; examiner blinded (OK!)
- lines 376-377: “subjected to plaque control twice a week after the root coverage intervention.” Explain detailing how it was done
- Was there a test to prove the cellular viability? As demonstrated by Fernandes et al. (2012), for example, DOI: 10.4028/www.scientific.net/KEM.493-494.37, who evaluated through three parameters the cell survival and integrity (XTT, Neutral Red Uptake (NR), and Crystal Violet Dye Exclusion (CVDE) tests). This fact may be fundamental to improve the understanding of the result found
- was there any sample calculation to determine seven animals with 56 defects?
- line 182: “From a total of 55 specimens, 49 were subjected to a histometric analysis”. Please, clarify the number 55 (7 animals with eight sites each = 56)
- Table 1: The analysis periods were two weeks and after 4 or 6 weeks? please, fix it there
- Where is the statistical analysis between time (2 and 6 weeks)?
- from 2.3.1 up to 2.3.5: improve the description in the text (they could be improved)
5. Discussion: this part started with new information: “this pilot preclinical investigation.” Is it a pilot study?
- lines 290-301: transfer from discussion to results
- lines 310-311: “although no statistically significant differences were found among groups…..”. The authors continued this paragraph, reporting that the values were "greater." Please, I suggest rewriting this part because there was no significant result (it could cause a misunderstanding to the reader)
- lines 331-332: it was not possible to affirm what is in these lines because there was no significant statistical result in the current study
6. CONCLUSION: I suggest inserting in the text that the results were similar (CAF alone had similar results) considering that cell therapy has a high cost typically and was not efficient.
- lines 338-339: “a lower inflammatory response when compared to the use of CAF and the CMX alone.” These are subjective findings, or could the authors clarify them using histomorphometric analysis?
Author Response
Response to Reviewer 2 Comments
We would like to thank the Reviewer 2 for his/her valuable comments.
Point 1: Abstract: “Dehiscence-type defects were created on seven dogs” and four groups “1) CAF plus the combination of CMX and GFP+ GMSCs, 2) CAF plus CMX 20 with autologous fibroblasts, 3) CAF plus CMX, 4) CAF alone” analyzed after “2 and 6 week-healing periods.”
Results - “differences weren’t statistically significant.”
The conclusion must be improved because it may cause a misunderstanding “effect of combining CMX and GMSCs with the CAF resulted in positive histological and clinical outcomes”; the values were not significant.
Response to Point 1: The conclusion of the abstract (“The effect of combining CMX and GMSCs with the CAF resulted in positive histological and clinical outcomes but the added value when comparing to the groups without cells was not significant”)has been modified: “the histometric and clinical results indicated a positive trend favouring the combination of CMX and GMSCs with the CAF when compared to the groups without cells although these differences were not statistically significant”.
INTRODUCTION: the introduction is direct and well-written (around 50% of the articles cited within five years)
OBJECTIVE: “evaluate the soft tissue regenerative potential of gingiva-derived MSCs (GMSCs) seeded in CMX when combined with the CAF.”
M&M: there was randomization; allocation to the tested interventions was concealed by using sealed envelopes; followed the ARRIVE guidelines; examiner blinded (OK!)
Point 2: lines 376-377: “subjected to plaque control twice a week after the root coverage intervention.” Explain detailing how it was done
Response to Point 2: It has been added: “The animals were fed using a granulated dog food, previously wetted in water, with individual bowls and free supply of water and subjected to plaque control twice a week after the root coverage intervention by using gauzes soaked with chlorhexidine 0.12% (Perio-Aid®, Dentaid, Cerdanyola, Spain).”
Point 3: Was there a test to prove the cellular viability? As demonstrated by Fernandes et al. (2012), for example, DOI: 10.4028/www.scientific.net/KEM.493-494.37, who evaluated through three parameters the cell survival and integrity (XTT, Neutral Red Uptake (NR), and Crystal Violet Dye Exclusion (CVDE) tests). This fact may be fundamental to improve the understanding of the result found
Response to Point 3: yes. To demonstrate cell viability as well as to count living cells, a Casy Cell Counter, based on the electrical current exclusion method for measuring cell viability of mammalian cell lines (validated in Lindl et al.2005) was employed. This is also indicated in the previous publication where the reader is referred to for additional information about cell processing (Santamaría et al. 2017). For improving the understanding of the results, this has been added to the Material and Methods section- Phase 3. Root coverage surgeries, Page 18):
“GMSCs and FBs were detached from the culture flask and the number of living cells counted with an automated cell counter (TTC model CASY®, 150 μm capillary, Roche Diagnostics, Basel, Switzerland) [36]”
Point 4: was there any sample calculation to determine seven animals with 56 defects?
Response to Point 4: no, there was not sample size calculation since this is a pilot study (no previous publication has evaluated the effect of mesenchymal stem cells seeded in a xenogeneic collagen in combination with the CAF for root coverage in animals). We selected the sample size according to the number of animals employed in other published preclinical studies in which a similar xenogeneic matrix is used for root coverage in dogs (e.g. Tal et al. 1996- 4 Dogs, Cha et al. 2017- 5 Mongrel dogs).
Tal H, Pitaru S, Moses O, Kozlovsky A. Collagen gel and membrane in guided tissue regeneration in periodontal fenestration defects in dogs. J Clin Periodontol. 1996 Jan;23(1):1-6. doi: 10.1111/j.1600-051x.1996.tb00496.x. PMID: 8636450.
Cha JK, Sun Y-K, Lee J-S, Choi S-H, Jung U-W. Root coverage using porcine collagen matrix with fibroblast growth factor-2: a pilot study in dogs. J Clin Periodontol 2017; 44: 96–103. doi: 10.1111/jcpe.12644).
Point 5: line 182: “From a total of 55 specimens, 49 were subjected to a histometric analysis”. Please, clarify the number 55 (7 animals with eight sites each = 56)
Response to Point 5: it was a writing error; modified to 56.
Point 6: Table 1: The analysis periods were two weeks and after 4 or 6 weeks? please, fix it there
Response to Point 6: It has been modified (6 weeks healing).
Point 7: Where is the statistical analysis between time (2 and 6 weeks)?
Response to Point 7: From an statistical point of view it would not be correct to analyse differences between 2 and 6 weeks since the results from both time periods are not provided by the same specimen (each sample is not analysed after 2 and 6 weeks but either 2 or 6 weeks) (“Each site was randomly assigned to one intervention and healing time, using a computer-generated randomization list stratified by hemi-mandible/maxilla, tooth, and healing time (IBM SPSS Statistics® V20. JM. Domenech;Page 15, Material and Methods-study design and randomization-line)
Point 8: from 2.3.1 up to 2.3.5: improve the description in the text (they could be improved)
Response to Point 8: done
2.3.1. Epithelial Length (M-aJE).
Although no significant differences among groups were found, the CMX group was slightly better in terms of mean epithelial length, with lower dimensions M-aJE at 2 (4.37 mm; SD=1.55) and 6 weeks (3.24 mm; SD=1.47). However, the CMX+GMSCs (5.32 mm; SD=1.49) at 2 weeks and the CMX+FBs group (4.57 mm; SD=1.19) at 6 weeks exhibited poorer outcomes than the other treatment groups (p>0.05) (Table 1).
2.3.2. Histological Root Coverage (cN-M)
The CAF group exhibited larger dimensions of histological root coverage (2.54; SD=0.92) than the CMX (0.61; SD=1.09) and the CMX+FBs (1.09; SD=0.72) groups at 2 weeks, with statistically significant differences between them (Table 1).
2.3.3. Length of the connective tissue attachment (aN-aJe)
Despite the lack of statistically significant differences between the groups, at 2 weeks, the CMX group showed better results than the other three groups since it achieved the greatest dimensions of connective tissue attachment (1.62; SD=3.44).. At 6 weeks, the four groups exhibited similar results, ranging from 1.30 to 1.69 mm (p>0.05) (Table 1).
2.3.4. Length of new-formed cementum (aN-cC)
At 2 weeks, the CMX+FBs group showed a higher dimension of new cementum formation, approximately 1 mm (1.09; SD=0.72). However, at 6 weeks, this group exhibited inferior results for this outcome variable (0.40; SD=0.65), in comparison with the other groups demonstrating cementum gains of at least 1 mm (p>0.05) (Table 1).
2.3.5. Dimensions of new bone (aN-Bc)
The only treatment group that showed mean bone gains at 2 weeks was the CMX+FBs group (p>0.05). However, at 6 weeks all groups demonstrated similar bone gain dimensions (Table 1)
Point 9: 5. Discussion: this part started with new information: “this pilot preclinical investigation.” Is it a pilot study?
Response to Point 9: Yes, this is a pilot study because no previous publication has evaluated the effect of mesenchymal stem cells seeded in a xenogeneic collagen in combination with the CAF for root coverage in animals. From this study, it can be obtained now the minimum expected difference between the groups and the standard deviation for sample size calculation in future preclinical studies.
We have add the word “pilot” in the design of the trial in the material and methods to clarify this point: “This study was designed as a preclinical split-mouth randomized controlled pilot study comparing four root coverage surgical protocols”.(Material and Methods- Study Design and Randomization-page 15)
Point 10: lines 290-301: transfer from discussion to results
Response to Point 10: it has been transferred to the results.
Point 11: lines 310-311: “although no statistically significant differences were found among groups…..”. The authors continued this paragraph, reporting that the values were "greater." Please, I suggest rewriting this part because there was no significant result (it could cause a misunderstanding to the reader)
Response to Point 11: It has been changed for “In terms of the clinical results, although no statistically significant differences were found among groups, the CMX+GMSCs groupshowed a trend towardsthe greatest RecRed values (1.42; SD=1.88 mm), the higher percentage of complete root coverage (57,1%) and the higher percentage of sites with ≥ 2 mm RecRed (57%) between baseline and 6 weeks(Table 2 and Appendix B).
Table 2 and Appendix B are referenced to enhance the reader to consult the results and significance of the results.
Table 2. Clinical variables at baseline and 2 and 6 weeks after the root coverage procedure.
Variable |
CAF (n=14) |
CAF+CMX (n=14) |
CAF+CMX+GMSCs (n=13) |
CAF+CMX+FBs (n=14) |
p-value |
Recession length in mm mean (SD) |
|||||
Baseline |
2.28 (0.97) |
1.60 (0.88) |
2.07 (1.26) |
1.89 (0.81) |
0.289† |
2 weeks |
0 |
0.57 (0.67) |
0.25 (0.41) |
0.14 (0.37) |
0.095† |
6 weeks |
0.57 (0.60) |
0.78 (0.69) |
0.57 (0.93) |
0.57 (0.83)) |
0.816† |
RecRed in mm mean (SD) |
|||||
Baseline-2 weeks |
2.85 (0.89) |
1.07 (1.39) |
1.75 (1.54) |
1.85 (1.02) |
0.085‡ |
Baseline-6 weeks |
1.14 (0.37) |
0.78 (0.90) |
1.42 (1.88) |
1.21 (0.63) |
0.749‡ |
Root coverage in % mean (SD) |
|||||
Baseline-2 weeks |
100 (0.0) |
46.42 (58,50) |
82.66 (28.90) |
92.85 (18.89) |
0.062† |
Baseline-6 weeks |
72.61 (28.34) |
45.23 (45.86) |
47.14 (94.28) |
75.00 (38.18) |
0.586† |
Specimens with CRC in % |
|||||
2 weeks |
100 |
42.9 |
60 |
85.7 |
0.077* |
6 weeks |
42.9 |
28.6 |
57.1 |
57.1 |
0.664* |
CAF, coronally advanced flap; CMX, collagen matrix; GMSCs, gingival mesenchymal stromal cells; FBs, fibroblasts; SD, standard deviation; RecRed, recession reduction; CRC, complete root coverage.
† Kruskal-Wallis test was used for quantitative variables with a non-normal distribution.
‡One-way ANOVA was used for quantitative variables with a normal distribution.
* Chi-Square test was used for categorical data.
Statistically significant differences were considered for p≤0.05.
Point 12: lines 331-332: it was not possible to affirm what is in these lines because there was no significant statistical result in the current study
Response to Point 12: lines 331-335 have been removed.
Point 13: CONCLUSION: I suggest inserting in the text that the results were similar (CAF alone had similar results) considering that cell therapy has a high cost typically and was not efficient.
Response to Point 13: it has been inserted
“the results were similar between the groups with no statistically significant differences”. Accordingto the results of the present study, for the time being, the use of MSCs is not cost-efficient.
Point 14: lines 338-339: “a lower inflammatory response when compared to the use of CAF and the CMX alone.” These are subjective findings, or could the authors clarify them using histomorphometric analysis?
Response to Point 14: We have not performed histomorphometric analysis but a dichotomous evaluation of presence/absence of intense inflammatory infiltrate in the connective tissue. Similarly to what it can be observed in figure 4., we found a clear presence of a denser inflammatory cell infiltration around the matrix in the samples treated with the CMX alone in comparison to the other three groups.
Please see the attachment "Manuscript with changes"

Round 2
Reviewer 2 Report
Dear authors,
thank you for the revision.